# Revisiting p38 Mitogen-Activated Protein Kinases (MAPK) in Inflammatory Arthritis: A Narrative of the Emergence of MAPK-Activated Protein Kinase Inhibitors (MK2i)

**DOI:** 10.3390/ph16091286

**Published:** 2023-09-12

**Authors:** Payal Ganguly, Tom Macleod, Chi Wong, Mark Harland, Dennis McGonagle

**Affiliations:** Leeds Institute of Rheumatic and Musculoskeletal Medicine, University of Leeds, Leeds LS9 7JT, UK

**Keywords:** p38-MAKP pathway, MK2 inhibitor, drug target, inflammation, inflammatory arthritis

## Abstract

The p38 mitogen-activated protein kinase (p38-MAPK) is a crucial signaling pathway closely involved in several physiological and cellular functions, including cell cycle, apoptosis, gene expression, and responses to stress stimuli. It also plays a central role in inflammation and immunity. Owing to disparate p38-MAPK functions, it has thus far formed an elusive drug target with failed clinical trials in inflammatory diseases due to challenges including hepatotoxicity, cardiac toxicity, lack of efficacy, and tachyphylaxis, which is a brief initial improvement with rapid disease rebound. To overcome these limitations, downstream antagonism of the p38 pathway with a MAPK-activated protein kinase (MAPKAPK, also known as MK2) blockade has demonstrated the potential to abrogate inflammation without the prior recognized toxicities. Such MK2 inhibition (MK2i) is associated with robust suppression of key pro-inflammatory cytokines, including TNFα and IL-6 and others in experimental systems and in vitro. Considering this recent evidence regarding MK2i in inflammatory arthritis, we revisit the p38-MAPK pathway and discuss the literature encompassing the challenges of p38 inhibitors with a focus on this pathway. We then highlight how novel MK2i strategies, although encouraging in the pre-clinical arena, may either show evidence for efficacy or the lack of efficacy in emergent human trials data from different disease settings.

## 1. Introduction

Mitogen-activated protein kinases (MAPKs) belong to a family of enzymes responsible for generating and coordinating several cellular responses via phosphorylation upon exposure to external stimuli. MAPKs are serine-threonine kinases involved in intracellular signaling in several cellular functions, including cell proliferation, differentiation, death, and survival [1]. All of the MAPK signaling cascades are initiated by extracellular cues leading to the activation of a particular MAPK followed by successive activation of MAPK kinase kinase (MAPKKK) and MAPK kinase (MAPKK) [2]. The MAPK pathway is typically activated by interactions with small GTPases and/or phosphorylation of protein kinases downstream from cell surface receptors and has been well reviewed before [1,3,4]. MAPKKK phosphorylates and activates MAPKK, which in turn activates MAPK by double phosphorylation [2]. On activation, MAPKs execute biological responses via protein function and gene expression. MAPKs are equipped with specific docking sites to ensure recognition of specific downstream targets [5].

Conventionally, there are three MAPK family subgroups: extracellular signal-regulated-kinases (ERKs); Jun-amino-terminal kinases (JNKs); and p38/stress-activated protein kinases (SAPKs) [2]. In mammals, it has been observed that the ERK-1 and -2 pathways are usually activated by mitogens such as phytohemagglutinin (PHA), pokeweed mitogen, or lipopolysaccharide (LPS) and are found to be upregulated in many human tumors. These are known as major regulators of the cell cycle, specifically the G1-to-S-phase transition [6]. On the other hand, JNK and p38 are activated by the presence of environmental or genotoxic stressors [7]. Discussing all three major pathways is beyond the scope of this article, but they are well reviewed elsewhere [8,9,10,11]. 

The p38-MAPKs were discovered as part of the screening process for identifying compounds that could regulate the production of tumor necrosis factor (TNFα) by LPS monocyte stimulation [12]. They are activated by a wide range of cellular stressors as well as in response to inflammatory cytokines [13]. Not only does the p38-MAPK pathway regulate inflammation, but it also regulates osteoclast differentiation and bone resorption via RANKL expression modulation [14]. Previously, it has been shown to be responsible for the production of inflammatory enzymes, including COX and iNOS [15], as well as the regulation of matrix metalloproteinase expression, including MMP2, MMP9, and MMP13 [16]. It is essential in the production and activation of pro-inflammatory cytokines and has attracted particular attention. There are four known members of the p38-MAPK family: p38α, p38β, p38γ, and p38δ, based on sequence homology, substrate specificities, and sensitivities to chemical inhibitors (Figure 1). All four members have about 60% amino acid sequence homology; however, they differ in their expression, substrate specificity, and sensitivities towards inhibitors [13]. Among them, p38α is the most researched and is expressed by almost all cell types, followed by p38β. The p38γ isoforms are largely expressed in skeletal muscle and p38δ in tissues in the kidneys, intestines, testis, and pancreas [17], but they have not been investigated as much as the α/β isoforms [18]. 

Mice without the γ/δ isoforms have been reported to be viable and have no apparent phenotype [19]. Risco et al. reported the involvement of p38δ and γ in the development of T lymphocytes [20]. In macrophages and dendritic cells that are key mediators of the inflammatory response, deletion of both p38δ and γ impairs the innate immunity response to LPS stimulation [19]. Studies on p38δ-deficient mice indicated that it played a vital role in neutrophil migration and in their recruitment at inflammatory sites in the lungs [21]. However, due to their low specificity, investigations of p38α and β still constitute the majority of knowledge of this pathway, so much so that p38α is usually referred to simply as ‘p38’ unless referring to another isoform. The p38α isoform has been shown to regulate cell proliferation and programmed cell death (or apoptosis), which are essential for normal functioning. Additionally, p38α-deficient cells were found to be more resistant to apoptosis, potentially indicating that it is a positive regulator of apoptosis [22]. Additionally, p38α has been indicated to directly phosphorylate over 100 proteins to regulate a plethora of functions, including transcription, mRNA stability and translation, metabolism, and the cell cycle [23,24,25].

Since its discovery, the understanding of the broader biological functions of p38 has increased significantly, but so has the complexity of this signaling pathway. For example, p38-MAPK may have an important role in maintaining the inflammatory status of endothelial cells via pro-inflammatory cytokines, whereas ERK1/2 of the MAPK pathway have been indicated to show anti-inflammatory stimuli on endothelial cells [26]. Interleukin-17 (IL-17) has been shown to promote p38-MAPK-dependent endothelial cell activation, enabling the recruitment of neutrophils at the sites of inflammation [27]. More recently, MAPK was indicated to play an important role in endothelial activation towards hematopoietic stem cell dysfunction within the bone marrow [28]. 

In addition to the several roles mentioned above, the MAPK pathway overall plays a very important role in toll-like receptor (TLR) signaling pathways. It is known that TLR activation is a result of cross-communication between multiple pathways, some of which potentially interact to fine-tune inflammatory responses [29]. In 2013, Peroval et al. demonstrated a ‘consistent role for the elements of MAPK pathway’ (p38, MEK, and JNK) in agonist-dependent regulation of cognate TLR mRNA levels [30]. Considering the established involvement of TLR in autoimmune- and inflammation-related diseases like lupus [31] and inflammatory arthritis [32], it is imperative to further dissect the p38-MAPK pathway.

Investigations, including small-molecule inhibitors of the p38-MAPK, primarily those targeting α and β [33], that have helped unravel the translational understanding of this pathway are discussed in detail in the following section. 

Previously, the classes of p38 inhibitors included the pyridinyl imidazole class of drugs that inhibited p38α and β isoforms but not the γ and δ isoforms [10]. Additionally, a study by Beardmore et al. in 2005 demonstrated that deletion of p38β in mice did not impact T cell production. In this model system, p38α was the major isoform involved in inflammation, and inhibitors for this target would not need activity against p38β [34]. It was also found that endogenous inhibitors, like MKP1 and MKP7, did not affect all isoforms, as they failed to inhibit p38γ as well as δ [35]. With respect to selectivity of p38 inhibitors, molecules like AMG-584 (by Amgen), BIRB-796 (by Boehringer Ingelheim), and Pamapinod (by Roche), all had higher selectivity for the α and β isoforms, with lesser selectivity for γ and even lower selectivity for δ isoform [36]. Thus, the majority of these inhibitors have focused on the p38α isoform, as indicated in the simplified figure below (Figure 2). 

p38-MAPK is one of the key regulators of pro-inflammatory cytokine production. Its phosphorylation, mainly of the α and β isoforms, leads to the activation and regulation of pro-inflammatory cytokines such as TNFα, interleukin-6 (IL-6), interleukin-1 (IL-1), IL-17, interleukin-18 (IL-18), and others. 

Cytokines play crucial roles in inflammatory conditions, and their expression has consistently been used as an indicator of the severity of inflammation. As previously mentioned, IL-1, IL-6, and TNFα are commonly expressed in inflammatory conditions such as inflammatory arthritis and have been extensively explored [37]. IL-1 is known to mediate autoimmune diseases [38] and its inhibition has demonstrated benefits to patients, not just those with inflammatory arthritis [39], but also patients with additional comorbidities like type 2 diabetes mellitus [40]. IL-6 is a pro-inflammatory cytokine and is among the most ubiquitously expressed cytokines in cancer [41], aging [42], and several conditions that involve inflammation [43]. Thus, inhibition of IL-6 also presents an opportunity as a therapeutic target in the treatment of auto-inflammatory diseases [44,45]. TNFα has long been identified as a key regulator of inflammatory responses and as a key target for relief from inflammatory diseases [46]. Recently, Loo and Bertrand have delineated the role of TNFα not only for inflammatory gene expression but also for its role in ‘inducing cell death, instigating inflammatory immune reactions and disease development’ [47].

IL-8, IL-12, IL-17, IL-18, and IL-23 are among the other cytokines that have also been found to play crucial roles in inflammatory diseases. Of note is IL-17, which is also a pro-inflammatory cytokine that promotes the development of diseases associated with immunity and inflammation [48]. In the rheumatology setting, it mediates cartilage and bone destruction via its action on osteoclasts (monocyte lineage cells), osteoblasts (bone cells) and synoviocytes (synovial cells) [49]. The importance of IL-17 in psoriatic disease and the spondyloarthropathies is clearly indicated by the outstanding success of anti-IL-17 therapeutics in this field [50,51]. Numerous other cytokines and their roles in inflammatory diseases have also emerged. These are discussed in detail by Kondo et al. [52], and those most relevant to inflammatory arthritis are outlined in Table 1 below.

p38-MAPKs are activated by disparate stress factors, including mechanical injury, oxidative stress, or pathogens, and they have been known to be involved in health in general. The downstream MK2 activation occurs via p38-MAPK, specifically their α and β isoforms, which bind to a basic docking motif in the C terminus of MK2 [57]. The primary role of p38-MAPK is to coordinate the molecular responses within the cell to stimuli associated with a diverse range of stressors [58]. These stressors may vary from changes in extracellular osmotic pressure [59] to pathogens [60] and thermal stress [61]. As the p38-MAPK pathway plays a significant role in stress-mediated inflammatory cytokine production (TNFα, IL-6, IL-1β, IL-18, and inflammatory mediator production, including COX-2), p38 inhibitors have been evaluated for inflammatory disease therapy [36,62,63,64]. In this article, we focus on the downstream p38-MAPK pathway antagonism, which may overcome the hurdles previously experienced in this therapeutic area.

## 2. Emergence of New Targets Downstream of p38-MAPK: MK2 Inhibition

The need for alternative p38 pathway antagonists emerged from p38-MAPK clinical trial failure. Schindler et al. have outlined several clinical trials using p38-MAPK inhibitors between 2002 and 2007 [33]. However, these inhibitors faced multifaceted challenges, including hepatotoxicity and cardiotoxicity, reflecting the wide functionality of the p38 pathway but also its lack of efficacy [62,65,66]. In some cases, even though early reduction of c-reactive protein (CRP) was observed, it was not sustained in spite of drug continuation [62,65]. This “tachyphylaxis” was observed in trials for Crohn’s disease [67] and, especially, rheumatoid arthritis (RA) [62,65]. In other reported cases, a lack of oral bioavailability was observed along with off-target effects and associated toxicity [68]. The need for new drug targets downstream of p38 thus emerged.

Considering its involvement in a range of physiological functions in health as well as in diseases, the p38-MK2 signaling pathway represents an interesting drug target [69]. MK2 is activated by the α and β isoforms of p38 by phosphorylation at Thr-222, Thr-334, and Ser-272 [61,70]. Activation of MK2 enhances the stability and translation of mRNA of the aforementioned pro-inflammatory cytokines [71]. This signaling axis is both downstream of receptors for inflammatory stimuli and upstream of the production of these pro-inflammatory molecules, thus allowing it to function as an amplifier of inflammation, as indicated by an increase in the expression of chemokines and cytokines [70,72]. Thus, the p38-MK2 signaling pathway has been associated with inflammatory disease states associated with cardiac conditions [73] and arthritis [10,74]. It has also been associated with cancer [75,76], gut aging [77], and pulmonary diseases related to acute lung injury and acute respiratory distress syndrome [78]. 

More recently, Soni et al. reviewed the significance of MK2 as a master regulator of RNA-binding proteins and its role in the regulation of transcriptional stability, particularly in tumor progression [79]. Beamer and Correa have outlined the importance of the p38-MK2 axis between neuro-inflammation and dysregulation in synaptic plasticity underlying cognitive impairments in neurological disorders [70]. Although it is anticipated that MK2 inhibition will have distinct advantages and less toxicity than pan p38 inhibitors, those observations call for careful evaluation in the clinical arena. Tristetaproline (TTP), an RNA binding protein, plays a critical role in regulating pro-inflammatory immune responses. The MK2 pathway phosphorylates TTP as a response to stress, and this leads to high RNA stability of stress-related expression by sequestering TTP from the AU-rich elements (ARE) (Figure 3) [80]. It acts as a driver in pathways triggered by DNA damage and has previously been expressed in a variety of cells, including endothelial cells [81] and smooth muscle cells [82]. In 1998, Ben-Levy and colleagues concluded that MK2 was required for the nuclear export of p38 and its eventual phosphorylation [83]. Since then, the role of MK2 has been explored in association with p38, largely in inflammation and inflammatory conditions in several tissues; that is further discussed below in health and disease.

A phase I placebo-controlled study of an MK2 inhibitor by Gordon et al. observed that the drug (ATI-450) was well tolerated with minor side effects, which supported its further investigation in inflammatory diseases [84]. Singh and colleagues compared the p38-MAPK inhibitors SB-203580 and BIRB-796 with MK2i PF-3644022 toxicity profiles and mechanism of action in vitro and in vivo [85]. They found that while both p38 and MK2 inhibitors exhibited strong anti-inflammatory properties by potently inhibiting levels of LPS-induced TNFα and IL-6 in human peripheral blood mononuclear cells (PBMCs), nuances were evident in their inhibitory profiles that would impact their anti-inflammatory potential. They outlined that BIRB-796 (the p38-MAPK inhibitor) led to a decrease in phosphorylation of mitogen- and stress-activated kinases (Msk1/2) with reduction of the anti-inflammatory cytokine IL-10 in the LPS-treated PBMCs [85]. However, this was not the case with PF-3644022 (the MK2i), which maintained the production of the anti-inflammatory cytokine IL-10 and indicated a dose-dependent reduction of IL-6. Additionally, evidence suggesting that MK2i would display lesser toxicity by avoiding participating in the feedback signaling loop and not activating the JNK pathway, potentially preventing tachyphylaxis was observed [85].

Previously, in a collagen-induced arthritis (CIA) model, MK2-deficient mice (MK2^−/−^ and MK2^−/+^) displayed reduced disease incidence in comparison to wild-type mice [86]. The study also indicated lower levels of TNFα and IL-6 production from the MK2-deficient mice than the wild-type mice [86]. Finally, MK2 gene deletion was beneficial and had a protective effect on the MK2-deficient mice when compared to that of wild-type mice [86]. This comparative study also predicted the targeting of MK2 inhibition for applications in RA. Later, Mourey et al. investigated the effect of PF-3644022, which they described as ‘a potent freely reversible ATP-competitive compound that inhibits MK2 activity’, using the U937 monocyte cell line and PBMCs, finding potent TNFα and IL-6 inhibition but not IL-1β or IL-8 inhibition [87].

More recently, Gordon et al. investigated ATI-450, an MK2 inhibitor, in a randomized, placebo-controlled phase 1 trial to evaluate its safety, tolerability, pharmacokinetics, and pharmacodynamics [84]. The subjects were given either a single ascending dose (SAD) or a multiple ascending dose (MAD) for up to seven days, and they observed a dose-dependent modulation of the target marker p-HSP27 as well as dose-dependent inhibition of production of TNFα, IL-6, IL-8, and IL-1β. They also found that p-HSP27 was generally well tolerated, with minor cases of dizziness, headaches, urinary tract infections, and constipation. However, these contraindications were not found to be dose-dependent [84].

In 2022, CC-99677 was introduced as an equivalent of the MK2 inhibitor to overcome the failures of p38-MAPK inhibitors, especially tachyphylaxis [88]. This employed a rare chloropyrimidine to bind to the sulfur of cysteine-140 in the ATP binding site via the nucleophilic aromatic substitution reaction (S_N_AR). The authors investigated the molecule and found that the cytokine suppression profile of CC-99677 was different from those observed for the known p38-MAPK inhibitors while avoiding tachyphylaxis. These included different inhibition patterns in the cytokines of IL-1β and monocyte chemoattractant protein 1 (MCP1), adding more evidence that MK2i could avoid the negative effects of p38-MAPK inhibitors [88].

The CC-99677 molecule was also used in rat experimental spondyloarthritis and was found to be efficacious, and 4–300 mg doses in healthy human volunteers indicated sustained TNFα levels with a favorable safety profile [88]. They further investigated CC-99677 in PBMCs of healthy donors and in patients diagnosed with AS in a first-of-its-type study with 37 donors randomly assigned to either placebo or the drug molecule [89]. They observed that the production of TNFα, IL-6, and IL-17 was inhibited in monocytes and macrophages in healthy donors and AS patients via a mRNA-destabilizing mechanism. Additionally, in the tachyphylaxis model, they observed a more differentiated pattern of sustained TNFα inhibition than that of the p38 inhibitors. Mechanistically, CC-99677 reduced TTP phosphorylation and accelerated the rate of decay of mRNA encoding inflammatory cytokines (TNFα, IL-6, and IL-17) in LPS-stimulated macrophages [89].

## 3. Inflammatory Arthritis and MK2 Inhibitors

Arthritic conditions account for a large number of cases of pain and the global burden of disease (GBD). RA alone contributed up to 18 million cases worldwide in 2019 [90] and its incidence is predicted to increase in the coming years [91]. As per the analysis of the 2019 GBD report, the population between 50 and 54 years of age had the highest incidence of RA, with over 10,000 cases of males and over 25,000 cases of females reporting the condition [91]. Thus, inflammatory arthritic conditions present key public health issues across the globe, with pain, stiffness, and inflammation in the affected joints as their main signs and symptoms. All of these lead to reduced movement and difficulties in performing daily tasks, which in turn reduce the quality of life (QOL) of the patients and raise the economic burden associated with the disease [92,93].

Current approaches towards treatment of inflammatory arthritic conditions include the use of non-steroidal anti-inflammatory drugs (NSAIDs), glucocorticoids, small-molecule disease-modifying anti-rheumatic drugs (DMARDs), and biological DMARDs or targeted synthetic DMARDs [94,95]. However, it must be noted that while remission of inflammatory arthritis is now achievable, many patients are still reported to not reach that stage. This can be attributed to patients with persistent inflammatory pathology and those with disease activity with non-inflammatory pathology [96].

Thus, a number of these inflammatory arthritic conditions, including RA, psoriatic arthritis (PsA), ankylosing spondylitis (AS), and juvenile inflammatory arthritis, have witnessed inconsistent patient relief and have a limited number of therapeutic options [97,98]. Recent evidence on MK2i molecules like CC-99677 (investigated for AS [89]) and ATI-450 (used successfully for phase I and phase IIa patients with RA [84,99]) has not highlighted a link to tachyphylaxis, a problem previously observed with p38-MAPK inhibitors. CC-99677 was found to display a linear pharmacokinetic profile and a high degree of target engagement, which resulted in sustained inhibition of the inflammatory cytokines [89]. These data present promising potential for this class of molecules for use in inflammatory and autoimmune conditions and merits further investigation.

This is especially relevant in 2023, when multiple Janus Kinase (JAK) inhibitors that have shown great promise as anti-inflammatory agents have earned a “black box” warning in the USA, caution against use in the EU region, and other global restrictions. The increased risk of serious heart-related events, cancer, blood clots, and death on treatment for certain chronic inflammatory conditions is what earned them the US Food and Drug Administration (FDA) warning [100,101] The first three JAK inhibitors to be approved by the FDA and European Medical Agency (EMA) were ruxolitinib (anti-JAK 1,2), tofacitinib (anti-JAK 1,3), and baricitinib (anti-JAK 1,2), which were also approved for their use in RA [102]. Evidence was collected from the World Health Organization’s (WHO’s) pharmacovigilance database called VigiBase, that contains over 20 million individual case safety reports (ICSRs). Analysis of these data indicated 126,815 ICSRs involved with JAK inhibitors. All three approved JAK inhibitors were associated with infectious adverse events, musculoskeletal and connective tissue disorders, embolism, and thrombosis, as well as with neoplasms [103]. Additionally, tofacitinib was also found to be associated with gastrointestinal perforation events. These concerns are underpinned by evidence for cardiovascular disease and venous thromboembolism, as well as cancer. This highlights the need to develop small molecules, or what are termed new targeted synthetic DMARDs. 

## 4. Challenges, Conclusions, and Future Directions

There have been reports outlining certain challenges faced by the first generation of ATP-competitive MK2 inhibitors. These include low solubility, poor cell permeability, and insufficient kinase selectivity [85]. During the process of manuscript finalization, a report emerged of a phase 2 study of the MK2i CC-99677 that was being investigated by BMS [89]. The phase 2 study reported that the study needed to be terminated due to a lack of short-term efficacy [104]. The study enrolled 167 subjects (18–65 years old) and then randomized them to receive oral MK2i candidate CC-99677 with either 150 mg, 60 mg, or placebo. At this juncture, despite the key positioning of the P38-MAPK pathway in immune regulation of cytokines, it appears that selective MK2 inhibition may be consigned to the junkyard of p38 pathway immuno-therapeutics, while JAK pathway antagonism blazes a trail in medicine. Nevertheless, the aforementioned positive AT1-450 phase IIa RA data by Gordon et.al in 2023 raises the possibility that MK2i may find relevance in inflammatory diseases [99].

New non-ATP-competitive MK2 inhibitors are now being formulated to overcome potential limitations [105]. Luber et al. presented preliminary data indicating that their MK2i (MMI-0100) that was delivered via inhalation displayed promising safety and tolerance in three different phase I clinical trials for patients with fibrotic and obstructive lung disease, with no adverse events reported in 75 subjects [106]. In another study, delivery of MK2i via nanopolyplexes (NPPs) enhanced cellular internalization, endosomal escape, and the half-life of MK2i for use in vascular graft hyperplasia [107]. This evidence suggests that the route of administration (ROA) and formulation chosen for MK2i may potentially impact the pharmacokinetics and pharmacodynamics (PK/PD) of the drugs.

Considering this, it is worth revisiting the p38-MAPK pathway for its downstream molecule, MK2, and its inhibition, especially as a therapeutic target for inflammatory arthritis. While a number of therapies exist for this condition, FDA warnings for JAK inhibitors indicate the need for more safe and efficacious drugs for administration to patients with any of the inflammatory arthritic conditions. Considering that recent information indicates MK2i to be superior to the traditional p38-MAPK inhibitors for inflammatory arthritis, specifically with respect to tachyphylaxis, which caused failure of the p38-MAPK inhibitors in several clinical trials, MK2 as a drug target and MK2i as drugs merit further biomedical and clinical research. 

In conclusion, new therapeutic options including MAPK and MK2 targeting are needed for the treatment of inflammatory arthritis. MK2i will possibly combat the problem of tachyphylaxis faced by the previous p38-MAPK inhibitors and continue its effect on reducing inflammatory cytokines over a long period of time. Additionally, evidence from recent studies in man is mixed but overall, MK2i could lower toxicity, provide higher efficacy, and the potential to reduce inflammation, at least in some disease settings. Indeed, given the link between JAKi and cardiovascular complications, it merits further investigation as an alternative therapy with potentially reduced cardiovascular risk. Finally, considering that p38-MAPK inhibition has resulted in improvement in other conditions such as tumors [108], atopic dermatitis [109], and neurological deficits [110], MK2i will likely be able to provide safe and efficacious therapy in these conditions as well.

## Figures and Tables

**Figure 1 pharmaceuticals-16-01286-f001:**
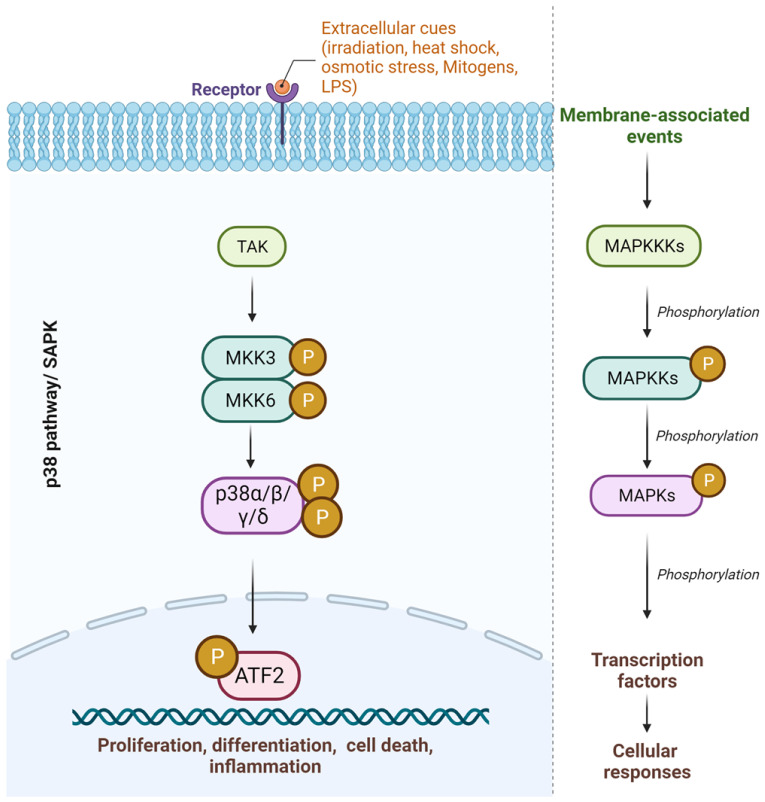
The p38-MAPK pathway overview.

**Figure 2 pharmaceuticals-16-01286-f002:**
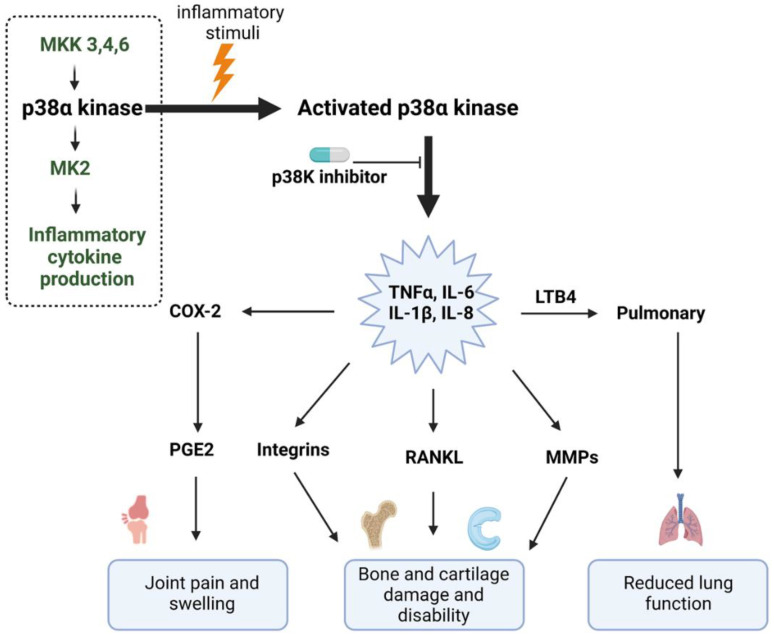
Role of p38 in inflammation and its inhibition and position of MK2 downstream of p38 (dotted box); figure was adapted from Schindler et al. [33].

**Figure 3 pharmaceuticals-16-01286-f003:**
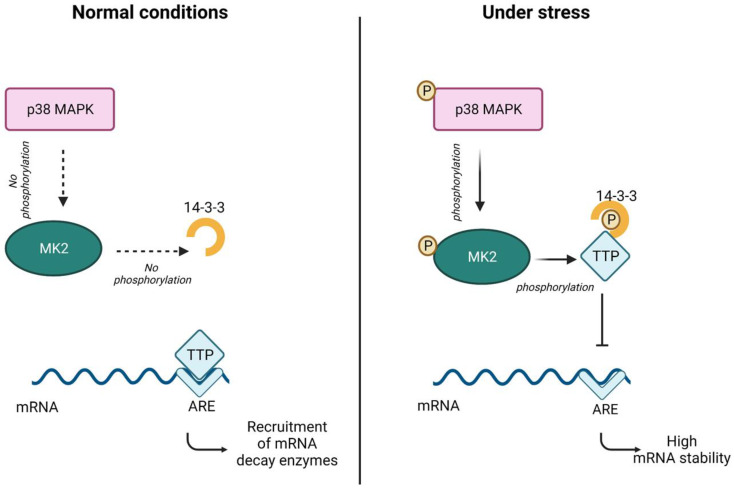
p38-MK2-TTP pathway upon exposure to stress stimuli. Under normal conditions, the p38-MAPK-MK2 pathway is not activated, and TTP is able to bind to ARE and recruit RNA decay enzymes. However, under stress, p38-MAPK is activated, which in turn activates MK2, and TTP is no longer able to bind to ARE. Figure was adapted from [80].

**Table 1 pharmaceuticals-16-01286-t001:** Cytokines regulated by p38-MAPK, their role in inflammation and some of the common drugs for these cytokine targets in inflammatory arthritis.

ILs Regulated	Role in Inflammation	References	Drugs Targeted for Inflammatory Arthritis
IL-1s, IL-1β	pro-inflammatory	[33,53]	Anakinra, Canakinumab, Rilonacept
IL-2	anti and pro-inflammatory	[33]	Baciliximab, Daclizumab for IL-2R
IL-3	pro-inflammatory	[33]	IL-3 inhibitor patent [54]
IL-6	pro-inflammatory	[7,33]	Tocilizuma, Sarilumab (IL-6R); Siltuximab
IL-8	pro-inflammatory	[33,55]	NA
IL-12	pro-inflammatory	[55]	Ustekinumab, Birankizumab
IL-17	pro-inflammatory	[56]	Secukinumab, Izekizumab
IL-18	pro-inflammatory	[57,58]	NA
IL-23p19	pro-inflammatory	[59]	Rizankizumab, Guselkumab

R—receptor, NA—not applicable.

## Data Availability

Not applicable.

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
