# Peer review of "Revisiting p38 Mitogen-Activated Protein Kinases (MAPK) in Inflammatory Arthritis: A Narrative of the Emergence of MAPK-Activated Protein Kinase Inhibitors (MK2i)"

_pharmaceuticals, 2023, doi:10.3390/ph16091286_

Round 1

Reviewer 1 Report

Excellent review on p38-MAPK pathway and the therapeutic potential of downstream molecule MK2 modulation as a therapeutic target for inflammatory arthritis, with up to date references.

General comments:

Table 1 and text, please consider including IL-18 given its role in inflammatory arthritis.

Fig. 2, it would be informative to include MAPK in endothelial activation.

Introduction – the role of MAPK in TLR signalling, in the context of inflammatory cytokines and arthritis.

General spacing issues – ie, Pg 5, line 158, byp38-MAPK.

Author Response

Please see document attached

Reviewer 2 Report

IN CURrENT PAPER  by TOPCU ET crucial signaling pathway p38-Mitogen activated protein kinase (p38-MAPK) and regarding MK2i in inflammatory arthritis, some revision of the p38-MAPK pathway and discussion the literature encompassing the challenges of p38 inhibitors with a focus on this pathway. Highlights in how novel MK2i may find a place in rheumatology, especially given the current concerns around JAK inhibition around cardiovascular diseases and cancer

comments good charts are depicted and research is sound, good work

good english uk

Author Response

Please see document attached

Reviewer 3 Report

Revisiting p38 mitogen activated protein kinases (MAPK) in 2 inflammatory arthritis: The emergence of MAPK-activated protein 3 kinase inhibitors (MK2i)

Impression: this is a concise review with issues that are relevant to clinicians and researchers. The followings are my suggestions that I believe would further improve the manuscript.

1. I suggest adding the phrase 'a narrative review' or something similar to the title to clear indicate the type of manuscript to the readership.

2. In the last section on future research and ongoing trials, I suggest summarizing the key data into a table. The table could include all the ongoing trials that the readers should look forward to, as well as therapeutic targets that may carry potential for future studies.

3. Table 1 felted too short and not very relevant. I suggest adding more data into the table. Perhaps data on the function and/or roles of each IL subtypes in inflammatory arthritis as a potential therapeutic target and what exisisting agent have already tackled these targets.

Author Response

Please see document attached
